# Deep Self-Dissimilarities as Powerful Visual Fingerprints

**Idan Kligvasser**
Technion–Israel Institute of Technology
kligvasser@campus.technion.ac.il

**Tamar Rott Shaham**
Technion–Israel Institute of Technology
stamarot@campus.technion.ac.il

**Yuval Bahat**
Technion–Israel Institute of Technology
yuval.bahat@campus.technion.ac.il

**Tomer Michaeli**
Technion–Israel Institute of Technology
tomer.m@ee.technion.ac.il

## Abstract

Features extracted from deep layers of classification networks are widely used as image descriptors. Here, we exploit an unexplored property of these features: their internal dissimilarity. While small image patches are known to have similar statistics across image scales, it turns out that the internal distribution of deep features varies distinctively between scales. We show how this deep self dissimilarity (DSD) property can be used as a powerful visual fingerprint. Particularly, we illustrate that full-reference and no-reference image quality measures derived from DSD are highly correlated with human preference. In addition, incorporating DSD as a loss function in training of image restoration networks, leads to results that are at least as photo-realistic as those obtained by GAN based methods, while not requiring adversarial training.

## 1 Introduction

Features extracted from deep layers of classification networks are widely used as powerful image descriptors. These features are known to capture high level semantics [44] as well as low-level textural cues [14, 31] and are thus exploited in numerous applications, including image enhancement [24, 18, 10, 47], synthesis [52, 44, 5, 19], and editing [15, 2, 25], both in the form of per-element similarity measures (*e.g.* the perceptual loss [15, 18] and LPIPS [51]) and for comparing internal image distributions (*e.g.* the style loss [15], contextual loss [32], and projected distribution loss [10]).

In this paper, we propose to exploit a surprisingly dominant property of deep features: the *dissimilarity* between their internal distributions at different image scales. As opposed to small patches in pixel space, which are known to exhibit similar characteristics across scales [53, 16], here we show that deep features tend to vary significantly for different scales of the same image. A glimpse into this phenomenon is provided in Fig. 1 for the VGG-19 network [46]. As can be seen, the network often outputs completely different classification results for the same image at different resolutions. We show that this behavior is also characteristic of earlier stages within the network, and can therefore serve as a powerful image fingerprint.

A naive approach to account for this phenomenon would be to aggregate deep feature descriptors from multiple image scales. For example, to measure discrepancy between two images, one could accumulate deep feature distribution distances, measured separately at different scales, as schematically illustrated in Fig. 5 (middle) for the case of two scales. However, here we propose a different approach, which as we show, is far more powerful as a visual fingerprint. Specifically, we present the *deep self-dissimilarity* (DSD) image descriptor, which captures differences between internal

35th Conference on Neural Information Processing Systems (NeurIPS 2021).

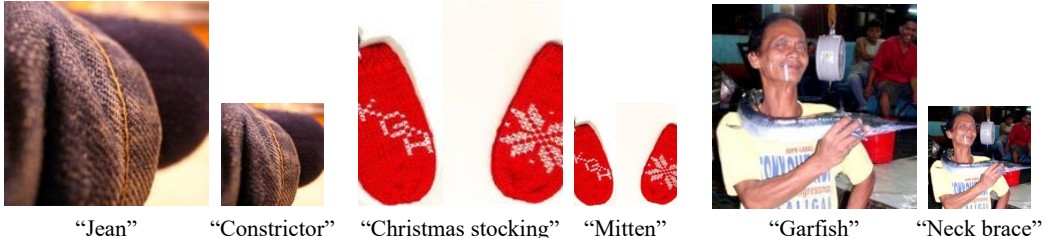

|          |              |                     |          |           |              |
|----------|--------------|---------------------|----------|-----------|--------------|
| "Jean"   | "Constrictor"| "Christmas stocking"| "Mitten" | "Garfish" | "Neck brace" |

Figure 1: **Class variations between scales.** The semantics captured by an image classifier (VGG-19 in this case [46]), strongly depends on the image resolution. This phenomenon also occurs at earlier classification layers; the deep features tend to vary significantly when changing the input's resolution. We show how these dissimilarities can serve to construct full-reference and no-reference image quality measures, as well as loss functions for image restoration.

feature distributions at different scales of an image. As we show, this descriptor can serve both as a full reference image fidelity metric (*i.e.* by comparing the DSD's of two images) and as a no-reference measure (quantifying the "naturalness" of an image according to its DSD). In both cases, our metrics are highly correlated with human perception, as we verify on user-annotated image quality assessment datasets. Furthermore, we demonstrate the effectiveness of DSD as a loss function for image restoration tasks. Our approach leads to high perceptual quality reconstructions that are at least comparable to GAN-based methods, while completely avoiding adversarial training.

## 2 Related work

**Deep features for similarity measures.** Using deep features of a pre-trained classification network for measuring fidelity between images, was first introduced in the context of network inversion and visualization [29, 45, 50]. This idea was quickly adopted as a perceptual similarity in tasks such as texture synthesis [14], style transfer [15, 18], super resolution [18, 24, 24], image inpainting [27] and more. Non-local (distribution based) variants of this per-element perceptual loss were also proposed. These include the style loss [15], the contextual loss [32], and the projected distribution loss [10]. These approaches compare between the internal deep-feature distributions of two inputs, at a single resolution. Here, we show that there is a more powerful way to use deep features: comparing their between-scale self-dissimilarities.

**Internal recurrence and self-similarity** Small image patches tend to recur within natural images, both in the same scale [8, 9, 6, 12], and across different scales [16]. Cross-scale recurrence has been used as a prior for various tasks, including super resolution [16, 13], denoising [54, 37], dehazing [4], blind deblurring [34, 3], and blind super resolution [33]. In this work we show that as opposed to image patches, deep features do not exhibit strong similarities across scales, implying that deep self-similarities cannot serve as a prior *e.g.* for restoration tasks. Nonetheless, we show that the *deep self dissimilarity pattern* of an image is meaningful, and can therefore serve as a powerful fingerprint.

## 3 The Self-Dissimilarity Between Deep Features at Different Scales

### 3.1 Deep self-dissimilarity

We characterize deep feature distributions through their second-order statistics. More concretely, we make use of Gram matrices of the channel activations (which are proportional to the empirical covariance of the channels),

$$[G_\ell(x)]_{i,j} = \frac{1}{WH} \sum_k [\phi_\ell^i(x)]_k \, [\phi_\ell^j(x)]_k. \tag{1}$$

Here $[\phi_\ell^i(x)]_k$ is the activation at spatial position $k$ within the $i$-th channel of the $\ell$-th layer of a network $\phi$ that is fed with the image $x$ at its input. $W$ and $H$ are the width and height of the this map.

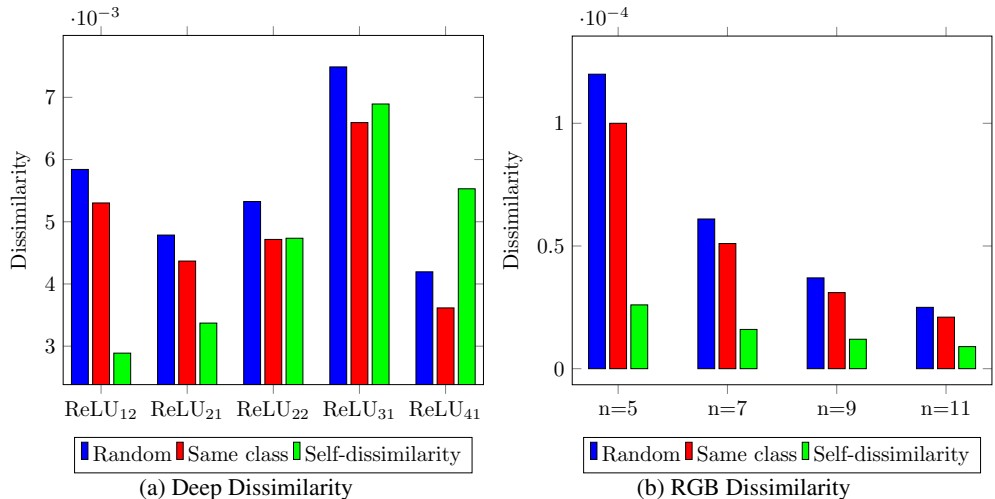

| (a) Deep Dissimilarity | (b) RGB Dissimilarity |

Figure 2: **Dissimilarity of image patches and deep features.** We compare the dissimilarities between images (blue & red) to the *self*-dissimilarity within images (green), both for deep features (a), and for small image patches (b). In contrast to image patches and shallow network layers, the large average self-dissimilarity levels (green) in deep network layers (right columns in (a)) are comparable to the cross-image dissimilarities (blue & red) in those layers. This shows that internal feature distributions in deep layers are not similar across image scales.

We define the deep self-dissimilarity (DSD) of an image $x$ as the difference between the $\ell$-th Gram matrices corresponding to two image scales, $x$ and $x{\downarrow_\alpha}$,

$$\mathrm{DSD}^\ell_\alpha(x) = G_\ell(x) - G_\ell(x{\downarrow_\alpha}). \tag{2}$$

In other words, $\mathrm{DSD}^\ell_\alpha(x)$ is a matrix whose $(i,j)$-th entry measures the extent to which channels $i$ and $j$ of the $\ell$-th layer are more correlated when feeding $x$ to the network, than when feeding $x{\downarrow_\alpha}$. We often choose the scaling factor $\alpha$ to be 2 or 4. We sometimes omit the $\alpha$ and $\ell$ subscripts for brevity.

## 3.2 Dissimilarity in RGB space vs. feature space

The strong tendency of small image patches (in pixel space) to recur across different scales of an image [53] is manifested in the similarity between distributions of patches across different image scales. If that was the case also for deep features, then DSD would always equal (approximately) the zero matrix, and thus would not constitute a unique fingerprint for images. However, as we show bellow, deep features behave very differently from small image patches in this respect.

We perform a comparison between deep feature self-dissimilarities and pixel-space self-dissimilarities for $\alpha = 4$. For the latter, we replace the deep feature Gram matrices by pixel-space patch Gram matrices,

$$[G_{\mathrm{RGB}}(x)]_{i,j} = \frac{1}{WH} \sum_k [x^i]_k \, [x^j]_k, \tag{3}$$

where $[x^i]_k$ is the $i$-th pixel in a column-stacked $n \times n$ patch, centered at the $k$-th pixel of the image $x$. $W$ and $H$ here correspond to the width and height of the image. For both deep features and pixel-space patches, we depict in Fig. 2: (i) the average $\ell_1$ distance between $G_\ell(x)$ and $G_\ell(x{\downarrow_4})$ over random images $x$ sampled from ImageNet [42] (green bars), (ii) the average $\ell_1$ distance between $G_\ell(x)$ and $G_\ell(y)$ over random *image pairs* $(x, y)$ from ImageNet (blue bars), and (iii) the average $\ell_1$ distance between $G_\ell(x)$ and $G_\ell(y)$ over random *image pairs* $(x, y)$ *sampled from the same class* in ImageNet (red bars). Distances are computed for several VGG layers $\ell$ and several patch sizes $n$.

As can be seen, the *self*-dissimilarity values (green) computed over image patches and shallow network layers are significantly lower than their cross-image counterparts (blue and red), in accordance with the well known cross-scale patch recurrence phenomenon in natural images. However, when it

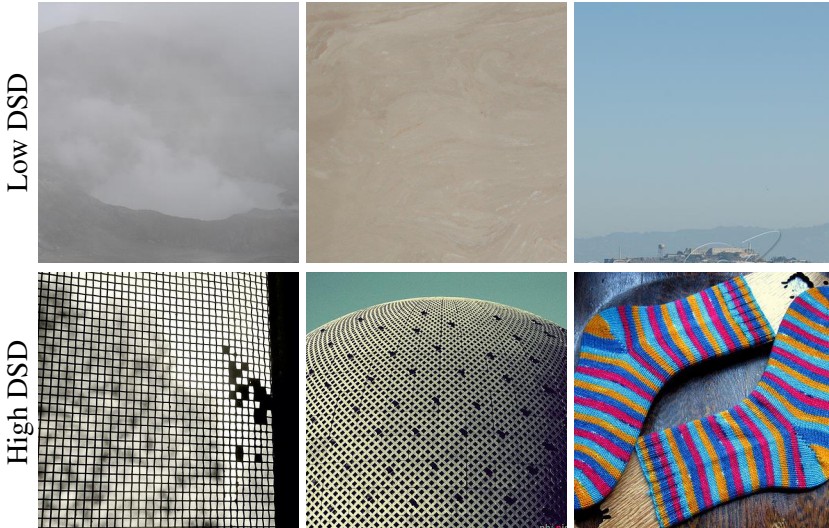

Figure 3: **Visualising DSD.** Images from the ImageNet validation set corresponding to lowest (top) and highest (bottom) DSD values. We observe that natural images with the lowest DSD tend to contain large smooth regions, whereas images with the largest DSD contain high frequency textures.

comes to deeper network layers (Fig. 2a), the average Gram dissimilarities between different scales of the same image are roughly as large as their cross-image counterparts, suggesting that deep features do not exhibit self similarity like pixel-space patches.

This conclusion is also supported by Fig. 6. Here, we optimized the RGB values of an image $x$ so as to minimize $\|\text{DSD}_4^\ell(x)\|_1$ for layer $\ell = \text{ReLU}_{31}$. As can be seen in the right image, too many (5K) gradient descent steps result in the image exhibiting unnatural artifacts. This illustrates again that natural images are in fact not characterized by small DSD values.

### 3.3 Visualizing DSD

What does the DSD fingerprint capture, then? We answer this question through several visualizations. Fig. 3 presents the images from the ImageNet validations set, which have the lowest and the highest DSD values layer at $\ell = \text{ReLU}_{31}$. As can be seen, images with higher DSD values (bottom row) typically contain finer details and textures, compared to low DSD value images (top row). This makes sense, as computing DSD involves downsampling the image, which results in erasing fine image details. This significantly changes the Gram matrix computed over deep features corresponding to the downsampled image, which in turn increases the difference between the two Gram matrices, computed in (2). Examining the statistics of DSD values across different classes in the ImageNet dataset reveals similar behaviour. For example the class "sandbar" has an average DSD of $2.1 \cdot 10^{-3} \pm 0.83 \cdot 10^{-3}$, while the average DSD of the class "window screen" is $13.5 \cdot 10^{-3} \pm 11.0 \cdot 10^{-3}$.

Next, we visualize the effect of modifying a specific content image $x_c$ such that its DSD fingerprint is as similar as possible to that of a reference image $x_r$. We do this once by minimizing $\|\text{DSD}_2(x_c) - \text{DSD}_2(x_r)\|_1$ and once by minimizing $\|\text{DSD}_2(x_c \downarrow_2) - \text{DSD}_2(x_r \downarrow_2)\|_1$. Namely, we try to match the images' deep self-dissimilarities between scales 1 and 0.5, and between scales 0.5 and 0.25. Here $\text{DSD}_2$ is calculated by aggregating the DSD matrices corresponding to VGG layers $\ell \in \{\text{ReLU}_{21}, \text{ReLU}_{22}, \text{ReLU}_{31}\}$. For comparison, we perform the same experiment, but with the style losses $\|G(x_c) - G(x_r)\|_1$, $\|G(x_c \downarrow_2) - G(x_r \downarrow_2)\|_1$, and $\|G(x_c \downarrow_4) - G(x_r \downarrow_4)\|_1$. The latter experiment can be thought of as a style pyramid, whereas the former as a style-differences pyramid (in analogy to the Gaussian and Laplacian pyramids). The results of the two experiments are presented in Fig. 4. As can be seen, while style dissimilarity (upper row) can be minimized by merely augmenting vague patterns from the reference image, matching the DSD fingerprints (bottom row) requires adding sharper, more detailed visual structures. This suggests that the DSD loss is a more intricate fingerprint, and can serve as a better loss for image quality assessment and for image restoration.

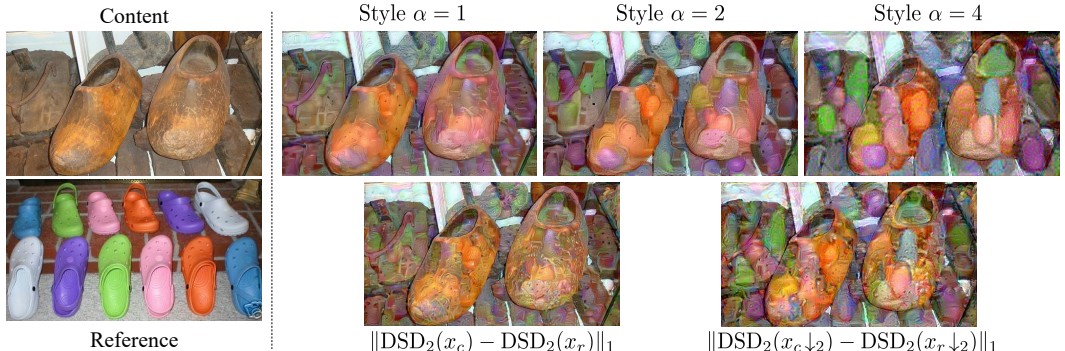

**Figure 4: The effect of changing the DSD of an image.** To visualize what the DSD fingerprint captures, we modify the content image to have a similar fingerprint to that of the reference image (bottom row). We do this for two pairs of scales (between full res. and $\alpha = 2$, and between $\alpha = 2$ and $\alpha = 4$). For comparison, we also perform the same experiment using the style loss for the three scales (top row). As can be seen, matching DSD fingerprints requires modifying finer details.

### 3.4 DSD as a full-reference measure

The DSD can be used to define a full-reference distortion measure simply by computing the $\ell_1$ distance between the DSD's of the two images we wish to compare,

$$\mathcal{L}_{\text{DSD}_\alpha}(x, y) = \|\text{DSD}_\alpha(x) - \text{DSD}_\alpha(y)\|_1. \tag{4}$$

This distance quantifies the extent to which the differences between the deep feature distributions at different scales of $x$ are similar to those of $y$ (see Fig. 5, right pane).

We evaluate this measure in a perceptual image quality assessment task. We use the PieAPP dataset [40], which contains 4800 pairs of images, where one image is the reference and the other is a distorted version of that reference. The dataset includes a variety of distortions, along with scores for their distortion level obtained from human ranking. We compare $\mathcal{L}_{\text{DSD}}$ with several other full-reference measures that are based on deep network features: the perceptual loss $\|\phi(x) - \phi(y)\|_1$, the style loss $\|G(x) - G(y)\|_1$, and the projected distribution loss (PDL) [10]. To isolate the effect of using multiple image scales in DSD, we also add a comparison to a multi-scale variant of the style loss, $\|G(x) - G(y)\|_1 + \|G(x\downarrow_2) - G(y\downarrow_2)\|_1$ (see schematic illustration in Fig. 5). Note that none of these metrics were specifically designed for (or even fine-tuned on) the PieAPP dataset.

We measured the correlation between each metric and the human preferences. Specifically, for every two distorted versions $x_a, x_b$ of the same reference image $x_r$, the PieAPP dataset [40] provides the probability that a human rater would prefer version $x_a$ over version $x_b$. We calculated the Pearson correlation between these probabilities and the loss differences $\mathcal{L}(x_b, x_r) - \mathcal{L}(x_a, x_r)$, for each of the full-reference losses $\mathcal{L}(\cdot, \cdot)$ we examined[1]. Table 1 presents the results for several different layers $\ell$ of the VGG network. Correlation with the LPIPS measure [51], which uses a different network, is also reported. Note that DSD values computed using the deeper network layers (two right columns) have the highest correlation with human evaluation scores, compared to all other measures. This illustrates that the DSD fingerprint can be used as an effective perceptual distortion measure.

### 3.5 DSD as a no-reference quality measure

The DSD fingerprint can also serve to define a no-reference image quality measure. Let us revisit Fig. 6, in which we optimized an image $x$ so as to minimize $\|\text{DSD}(x)\|_1$. Here, the original image is a super-resolution result produced by the ESRGAN method [48]. As already mentioned, after 5K gradient descent steps the optimized image contains unnatural artifacts. However, note that an interesting phenomenon occurs if we stop the optimization after only 500 steps. In that case, the optimized image is sharper and more naturally looking than the original ESRGAN result. This

---

[1]The Pearson correlation measures the extent to which the loss difference is a monotonic function of the preference probability.

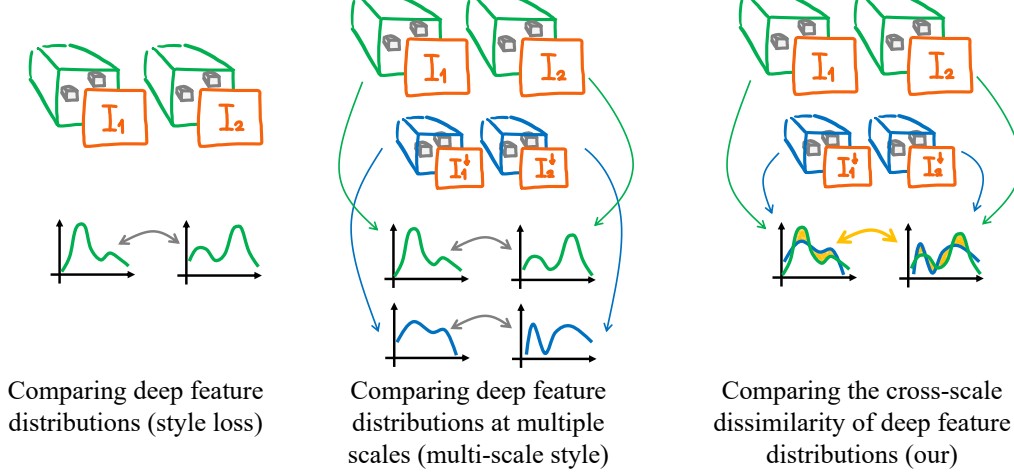

| Comparing deep feature distributions (style loss) | Comparing deep feature distributions at multiple scales (multi-scale style) | Comparing the cross-scale dissimilarity of deep feature distributions (our) |

Figure 5: **Deep self-dissimilarities as a full reference distortion measure.** As opposed to the standard style loss, which compares two images according to the distance between their Gram matrices at a single image scale (leftmost), we wish to exploit the *dissimilarity* between different image scales. A naive approach is to sum the style loss across scales (middle). However, we show that comparing the self-dissimilarity of deep features between image scales (rightmost) is a much more powerful metric.

| Method | $\text{ReLU}_{21}$ | $\text{ReLU}_{31}$ | $\text{ReLU}_{41}$ | $\text{ReLU}_{51}$ |
|---|---|---|---|---|
| Perceptual | 0.448 | 0.449 | 0.461 | 0.630 |
| PDL | 0.622 | 0.636 | 0.620 | 0.661 |
| Style | **0.645** | **0.714** | 0.711 | 0.706 |
| Multi-scale Style | **0.645** | 0.698 | 0.709 | 0.714 |
| $\mathcal{L}_{\text{DSD}_2}$ (ours) | 0.463 | 0.608 | **0.769** | **0.761** |
| $\mathcal{L}_{\text{DSD}_4}$ (ours) | 0.570 | 0.667 | 0.607 | 0.600 |
| LPIPS | | | 0.641 | |

Table 1: **DSD as a full reference measure.** We calculate the Pearson correlation between full-reference image distortion measures and human evaluation scores using the PieAPP dataset [40]. As can be seen, the DSD calculated on deep network layers achieves the highest correlation.

suggests that for this particular scene, there is a specific optimal $\|\text{DSD}(x)\|_1$ value, which is in between that of the original image, and the 5K minimization result.

Following this observation, we propose to use DSD for computing a no-reference image quality score. To asses the quality of a potentially degraded image $y$, we would like to measure the difference between $\|\text{DSD}_\alpha(y)\|_1$ and $\|\text{DSD}_\alpha(x)\|_1$, where $x$ is the degradation-free version of $y$. Obviously, in the no-reference case we do not have access to $x$. One could conceive replacing $\|\text{DSD}_\alpha(x)\|_1$ by its average value for natural images. The problem with this approach is that different natural images have very different $\|\text{DSD}_\alpha(x)\|_1$ values, as evident from the histogram in Fig. 7. A different approach would be to train a regression model to predict $\|\text{DSD}_\alpha(x)\|_1$ from $y$. However, we do not want to restrict our measure to specific degradations, and thus would like to avoid using datasets of degraded images for training. Our solution is to train a light-weight regression model $\psi$ to predict $\|\text{DSD}_\alpha(x)\|_1$ from the downsampled clean input, $x\downarrow_\alpha$. At test time, we input to our model the downsampled degraded image, $y\downarrow_\alpha$, and use the model's output as our estimate for $\|\text{DSD}_\alpha(x)\|_1$. The reasoning behind this choice, is that for many types of degradations $y\downarrow_\alpha$ and $x\downarrow_\alpha$ are quite similar (*e.g.* noise, blur, compression artifacts, etc.). We train the regression network $\psi$ (we use a convolutional network with 12 residual blocks [17]) on the BSD training set [30] containing 400 clean images, using the mean square error loss. Our no-reference measure is therefore given by

$$\mathcal{Q}_{\text{DSD}_\alpha}(y) = |\psi(y\downarrow_\alpha) - \|\text{DSD}_\alpha(y)\|_1|. \tag{5}$$

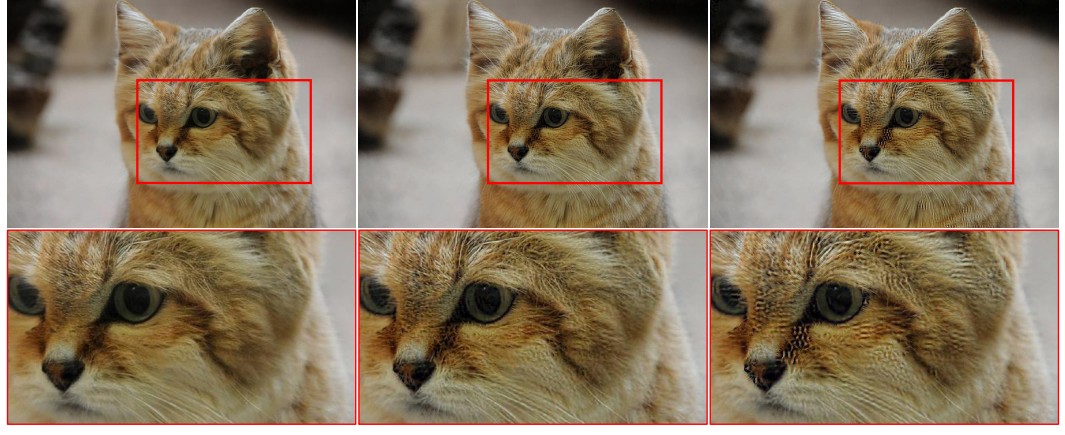

| DSD is too high (ESRGAN) | Optimal DSD (500 iterations) | DSD is too low (5K iterations) |

Figure 6: **Optimal deep self-dissimilarity.** We minimize $\|\mathrm{DSD}_4(x)\|_1$ starting from an output by the ESRGAN [48] method (left), having $\|\mathrm{DSD}_4(x)\|_1 = 6.3 \cdot 10^{-3}$. The result (right) after 5K iterations (with $\|\mathrm{DSD}_4(x)\|_1$ dropping to $1.2 \cdot 10^{-3}$) contains undesired artifacts. However, stopping the optimization earlier (after 500 steps, when $\|\mathrm{DSD}_4(x)\|_1 = 2.2 \cdot 10^{-3}$) yields a sharper and more natural looking image (middle), compared to the original ESRGAN output.

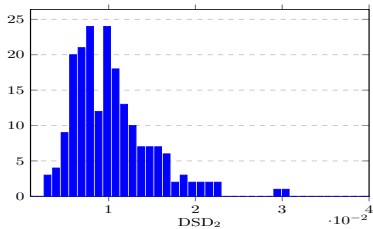

| Method | PCC |
|---|---|
| NIQE | 0.223 |
| Ma | 0.284 |
| BRISQUE | 0.363 |
| $\mathrm{DSD}_2$ | 0.330 |
| $\mathrm{DSD}_4$ | **0.444** |

Figure 7: **DSD distribution.** Histogram of $\|\mathrm{DSD}_2(x)\|_1$ calculated over 200 natural images from the PieAPP dataset [40]. DSD values strongly vary across different images.

Table 2: **DSD as a no-reference measure.** We report the Pearson correlation between human evaluation scores from [40] and various no-reference metrics, including our $\mathrm{DSD}_\alpha$ measure, with $\alpha = 2$ and $\alpha = 4$. As can be seen, $\mathrm{DSD}_4$ significantly outperforms the other quality measures.

We compare our proposed measure to the leading existing no-reference image quality measures, including the naturalness image quality evaluator (NIQE) [36], the blind/referenceless image spatial quality evaluator (BRISQUE) [35] and the score proposed by Ma et al. [28]. Similarly to the full-reference case, we compute the Pearson correlation between the human evaluation scores from [40] and the evaluated scores, and present the results in Tab. 2. $\mathrm{DSD}_4$ significantly outperforms the competitors in terms of correlation with human assessment.

## 4 Experiments

We use the DSD fingerprint for the tasks of single image super-resolution (SR) and motion debluring. In both cases our goal is to recover a clean image $x$ from its degraded observation $y$. To this end, we train a restoration network on pairs of training examples $\{x, y\}$ by minimizing the DSD-based full reference loss term (4) between the restored image $\hat{x}$ and its ground-truth counterpart $x$. We combine our loss with two other popular loss terms, as common in restoration methods (*e.g.* [24, 48]),

$$\mathcal{L}(x, \hat{x}) = \mathcal{L}_{\mathrm{per}}(x, \hat{x}) + \lambda_{\mathrm{rec}} \cdot \mathcal{L}_{\mathrm{rec}}(x, \hat{x}) + \lambda_{\mathrm{DSD}} \cdot \mathcal{L}_{\mathrm{DSD}}(x, \hat{x}). \quad (6)$$

Here $\mathcal{L}_{\mathrm{rec}}(x, \hat{x}) = \|x - \hat{x}\|_1$, $\mathcal{L}_{\mathrm{per}} = \|\phi_\ell(\hat{x}) - \phi_\ell(x)\|_1$ is the perceptual loss computed over layer $\mathrm{Conv}_{54}$ of VGG, and $\lambda_{\mathrm{rec}}, \lambda_{\mathrm{DSD}}$ are weighting coefficients. For $\mathcal{L}_{\mathrm{DSD}}$ we use layers $\ell \in \{\mathrm{ReLU}_{21}, \mathrm{ReLU}_{22}, \mathrm{ReLU}_{31}\}$ (as in Sec. 3.3) and $\alpha = 2$. We train our networks for 300K epochs using the Adam optimizer with a batch size of 16 and an initial learning-rate of $2 \cdot 10^{-4}$, which is halved after 90K, 180K and 270K steps. See Supplementary Material (SM) for full training details.

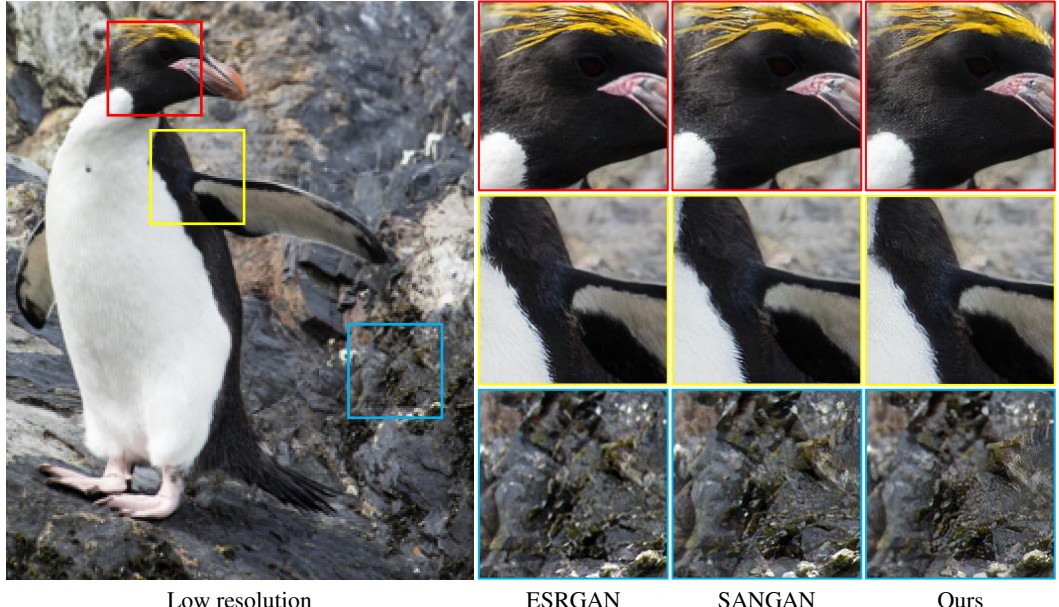

| Low resolution | ESRGAN | SANGAN | Ours |

Figure 8: **DSD for** $4\times$ **super resolution.** We compare our method to the state-of-the-art ESR-GAN [48] and SANGAN [21]. As can be seen, our approach leads to sharper and more photo-realistic images comparing to the GAN based methods, although we use no adversarial training.

## 4.1   Single Image Super Resolution

Here our goal is to predict a high-resolution (HR) image $x$ from its low-resolution (LR) version $y$. We focus on $4\times$ SR, where the LR image corresponds to bicubic downsampling of the HR image. Training is done over 800 LR-HR image pairs from the DIVK2K dataset [1]. Our SR network utilizes the xSRResNet architecture [22] consisting of 10 residual blocks.

Figure 8 presents a qualitative comparison with ESRGAN [48] and SANGAN [21], two state-of-the-art GAN-based methods that target perceptual quality. Note that although our method does not require any adversarial training, it manages to restore finer image details, and to produce more realistic textures. Please refer to the SM for many additional visual examples.

We next follow [7] and quantitatively evaluate the performance of our method by reporting perceptual quality (calculated using NIQE [36], lower is better) and image distortion (using SSIM [49], higher is better). In Fig. 9a we compare our model (red dot) with the leading SR methods EDSR [26], VDSR [20], SRResNet [24], xSRResNet [22], Deng [11], ESRGAN [48], SRGAN [24], ENET [43] and SinGAN [41] (black dots). We also compare variants of our model, trained by replacing $\mathcal{L}_{\mathrm{DSD}}$ with other deep internal-distribution based losses (blue dots): the projected distribution loss (PDL) [10] and the style loss (STY) [15]. Finally, we present results for our model trained using only $\mathcal{L}_{\mathrm{rec}}$ and $\mathcal{L}_{\mathrm{per}}$, or only $\mathcal{L}_{\mathrm{rec}}$ (indicated by "per" and empty subscripts respectively). All scores are calculated over the BSD100 test set [30]. As can be seen, our method is among the best in terms of perceptual quality, while maintaining relatively low distortion. Note that achieving such perceptual quality has been possible to date only with GAN based methods. Our method achieves such performance without the need of adversarial training, which is known to be unstable and challenging in practice.

To further investigate the contribution of the DSD loss, we perform quantitative evaluation of our method when $\mathcal{L}_{\mathrm{DSD}}$ is replaced by other loss terms that measure distances between deep feature distributions, including the contextual loss (CX) [32], projected distribution loss (PDL) [10], style loss [15] and the multi-scale style loss (MSS) defined in Sec. 3.4. For each of these loss terms, we adjust the value of $\lambda_{\mathrm{DSD}}$ so as to make the perceptual and feature distribution loss terms equal on average. The results are presented in Tab. 3, which reports PSNR, LPIPS [51] and NIQE [36]. In each experiment we calculate all losses using the same VGG layers (indicated at the top of each column), as well as the same training procedure and network architecture (here we use a shallow xSRResNet [22] with 5 residual blocks). In terms of perceptual quality (NIQE and LPIPS), the model

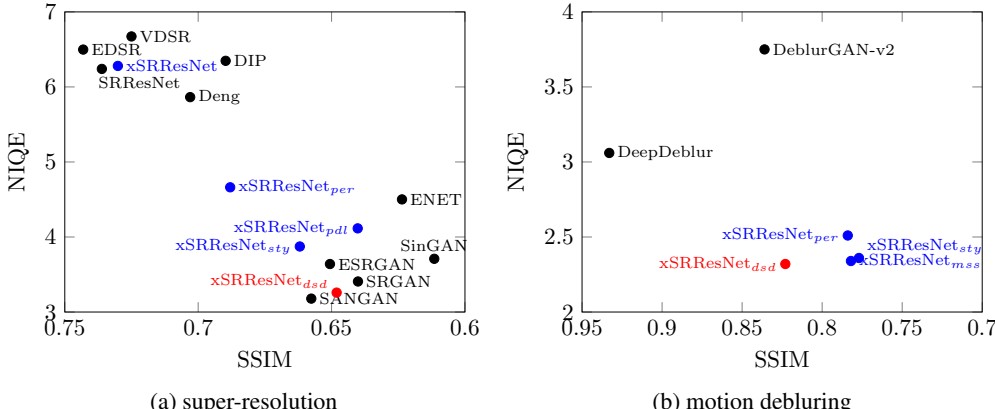

|                           | (a) super-resolution | (b) motion debluring |
|---------------------------|:--------------------:|:--------------------:|

Figure 9: **Perception-distortion evaluation.** We report perceptual quality (NIQE, lower is better) and image distortion (SSIM, higher is better) for the tasks of (a) super-resolution and (b) motion debluring. We compare our models (red), existing state-of-the-art algorithms (black) and variants of our model trained by replacing $\mathcal{L}_{\text{DSD}}$ with other deep feature distribution loss terms (blue). For SR, our method is among the best in perceptual quality, while requiring no GAN training. For motion debluring, our model improves over existing methods in perceptual quality by a large margin, while maintaining a relatively low distortion.

| Method | $\text{ReLU}_{21}$ | $\text{ReLU}_{22}$ | $\text{ReLU}_{31}$ |
|--------|--------------------|--------------------|--------------------|
| CX | **25.35** / 0.251 / 7.87 | **25.77** / 0.253 / 6.02 | **25.69** / 0.246 / 5.98 |
| PDL | 25.21 / 0.208 / 5.80 | 25.22 / 0.212 / 4.33 | 25.25 / 0.211 / 4.50 |
| Style | 24.93 / 0.197 / 3.64 | 24.84 / 0.188 / **3.73** | 24.66 / **0.190** / 4.20 |
| MSS | 24.77 / 0.197 / 3.86 | 24.72 / 0.190 / 4.01 | 24.87 / 0.196 / **3.87** |
| DSD (Ours) | 24.88 / **0.194** / **3.49** | 25.11 / **0.187** / 3.78 | 24.99 / **0.190** / 4.06 |

Table 3: **Comparison with other feature distribution losses in $4\times$ SR.** PSNR / LPIPS / NIQE scores for different loss functions utilizing different VGG layers. DSD based models (bottom row) perform among the best in terms of perceptual quality (NIQE and LPIPS).

trained with our DSD loss (bottom row) is among the best for all examined activation layers. This is also supported by Fig. 10, which shows a visual comparison between the methods. Our result contains less grid artifacts compared to the other losses (exemplified here for layer $\text{ReLU}_{22}$).

## 4.2 Single Image Motion Deblurring

Here we aim to recover a sharp, blur free image $x$ from a blurry image $y$. We use the same loss function, training protocol and architecture (xSRResNet [21] with 10 res-blocks) as in our SR experiments. Training is done using the REDS dataset [39] consisting of $30,000$ image pairs $\{x, y\}$ from 300 different scenes. Figure 11 presents an example result of our motion deblurring, compared with the state-of-the-art DeblurGAN-v2 [23] and a variant of our method trained with the style loss [15] instead of $\mathcal{L}_{\text{DSD}}$. The result using the DSD loss is sharper and contains no ghosting artifacts. Please see many more results in the SM.

This observation is further supported quantitatively. In Fig. 9b we report perceptual quality and distortion (using the same measures as in Fig. 9a) over 100 random images from the REDS validation set. We compare our method (red dot) with the state-of-the-art DeblurGAN-v2 and DeepDeblur[2] [38] (black dots), as well as with three variants of our method (blue dots) trained by replacing $\mathcal{L}_{\text{DSD}}$ with either the style loss (STY) [15], the multi-scale style loss (MSS) (from Sec. 3.4), and when omitting $\mathcal{L}_{\text{DSD}}$ and keeping only $\mathcal{L}_{\text{rec}}$ and $\mathcal{L}_{\text{per}}$ (denoted by a subscript "per"). Here as well, the perception-distortion plot indicates that our DSD-based model obtains the best perceptual quality (significantly better than most of the competition), while exhibiting low distortion.

---

[2]An official newer version available at: https://github.com/SeungjunNah/DeepDeblur-PyTorch

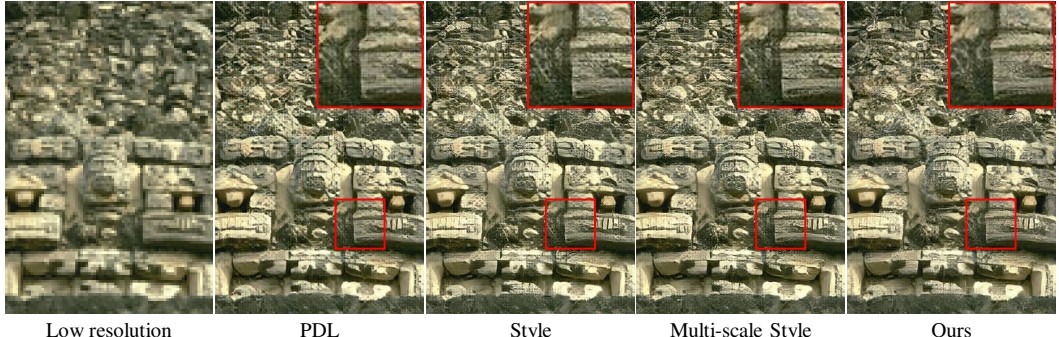

| Low resolution | PDL | Style | Multi-scale Style | Ours |

Figure 10: **Looking closer into the effect of DSD.** Substituting $\mathcal{L}_{\mathrm{DSD}}$ with other loss terms that are based on feature distributions results in undesired grid artifacts (middle images). These artifacts get significantly reduced when using the DSD loss term (right).

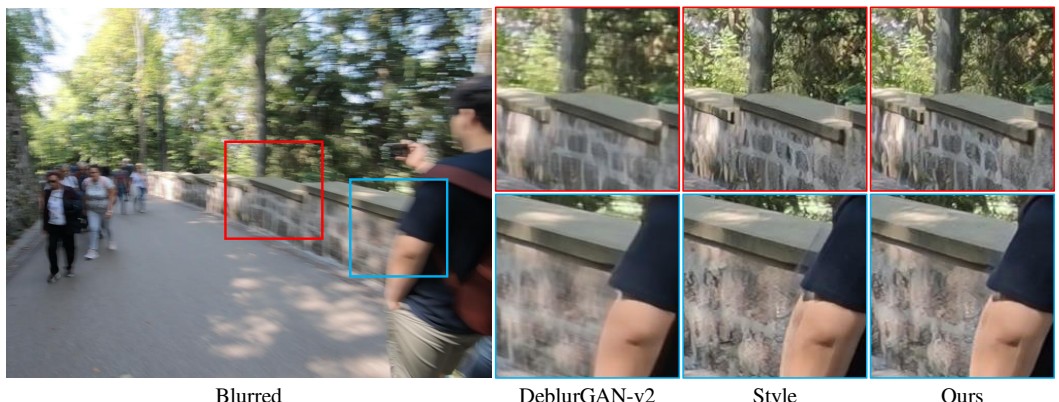

| Blurred | DeblurGAN-v2 | Style | Ours |

Figure 11: **DSD for motion debluring**. We visually compare our result with DeblurGAN-v2 and with a variant of our method where the DSD loss is replaced by the style loss. As can be seen, DSD achieves better visual quality, leading to a sharper, more natural appearing result.

## 5 Conclusion

Deep features corresponding to different image scales exhibit meaningful dissimilarity. We prove this to be a powerful image fingerprint, highly correlated with human preference in both full-reference and no-reference image quality assessment, and leading to a GAN-like highly photo-realistic image restoration (while avoiding unstable adversarial training). We believe that deep self-dissimilarity can benefit additional image restoration and manipulation tasks.

**Acknowledgments** This research was supported by the Israel Science Foundation (grant 852/17) and by the Technion Ollendorff Minerva Center.

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
