# Deep Self-Dissimilarities as Powerful Visual Fingerprints
# Supplementary Material

## 1 Experimental Setting

Tables 4 and 5 provide descriptions of the network architectures we use in each of the experiments. In Tab. 4 both networks consist of 10 residual blocks (as in [2]). We use leaky ReLU with slope 0.1, xUnit as activations [2] and pixelshuffle (PS) for the up-sampling.

| Super-resolution | Motion-deblurring |
|---|---|
| RGB image $x \in \mathbb{R}^{h \times w \times 3}$ | RGB image $x \in \mathbb{R}^{h \times w \times 3}$ |
| $5 \times 5$, stride=1 conv. 64 lReLU | $5 \times 5$, stride=1 conv. 64 lReLU |
| xResBlock, 64 | xResBlock, 64 |
| xResBlock, 64 | xResBlock, 64 |
| ⋮ | ⋮ |
| xResBlock, 64 | xResBlock, 64 |
| $3 \times 3$, stride=1 conv. 256, PS, lReLU | $3 \times 3$, stride=1 conv. 64, lReLU |
| $3 \times 3$, stride=1 conv. 256, PS, lReLU | $5 \times 5$, stride=1 conv. 3 |
| $3 \times 3$, stride=1 conv. 64, lReLU | |
| $5 \times 5$, stride=1 conv. 3, +up-sample(x) | |

Table 4: **xSRResNet network architecture.**

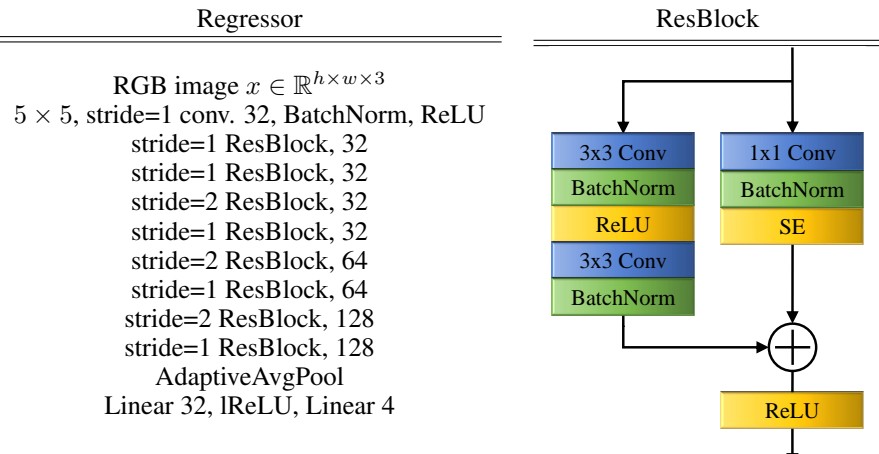

Table 5: **Regression network of Sec. 3.5.** The network $\psi$ predicts 4 different $\|\text{DSD}_\alpha^\ell\|_1$ values $\{\psi_\ell\}_{l=1}^4$, corresponding to VGG layers $\ell \in \{\text{ReLU}_{12}, \text{ReLU}_{21}, \text{ReLU}_{22}, \text{ReLU}_{31}\}$. Equation (5) in the main text is taken to be the average of the 4 layers, $\mathcal{Q}_{\text{DSD}_\alpha}(y) = \frac{1}{4}\sum_\ell |\psi_\ell(y\downarrow_\alpha) - \|\text{DSD}_\alpha^\ell(y)\|_1 |$. We train two regression networks, predicting the DSD values corresponding to $\alpha = 2$ and $\alpha = 4$. For the residual block we use the squeeze and excitation module [1].

## 2 Additional visual results

We provide additional visualization results for DSD as in Sec. 3.3 in the main text. We also include additional visual results for the task of super-resolution and motion-deblurring. Specifically, we have the following:

1. Figure 11 shows an additional visualization of the effect of the DSD loss. As can be seen, matching DSD fingerprints requires modifying finer details comparing to style loss.

2. Figure 12 presents additional visualizations of $\|\text{DSD}_4(x)\|_1$ minimization. The original image is a super-resolution result produced by the ESRGAN method [6]. After 5K gradient descent steps the optimized image contains unnatural artifacts. However, the results corresponding to intermediate $\|\text{DSD}_4(x)\|_1$ values, obtained after only 500 steps, are sharper and more naturally looking compared to both the initial (ESRGAN) and the over-optimized images. This support our conclusion that natural images are not characterized by small DSD values, and that there is an optimal DSD value that, which is in between that of the original image, and the 5K minimization result.

3. Figure 13 shows a visual comparison between different losses that utilize feature distributions. All methods in this figure share the same network architecture, which consists of only 5 residual blocks. As can been seen, our results contain less grid artifacts compared to the other methods.

4. Figures 14 and 15 show visual super-resolution comparisons. We compare our method to the state-of-the-art SRGAN [4] and ESRGAN [6] methods. Note that our approach leads to sharp and photo-realistic image restorations, comparable with the results by the GAN based methods, although we use no adversarial training. Note also that our neural network has fewer parameters than SRGAN and much fewer than ESRGAN (1.1M, 1.54M and 16.7M, respectively).

5. Figure 16 shows additional motion-debluring comparisons. We compare our results with DeepDeblur [5], DeblurGAN-v2 [3]. As can be seen, DSD achieves better visual quality, leading to sharper, and more natural looking results. Note also that our neural network has fewer parameters than DeepDeblur and much fewer than DeblurGAN-v2 (0.81M, 11.72M and 60.93M, respectively).

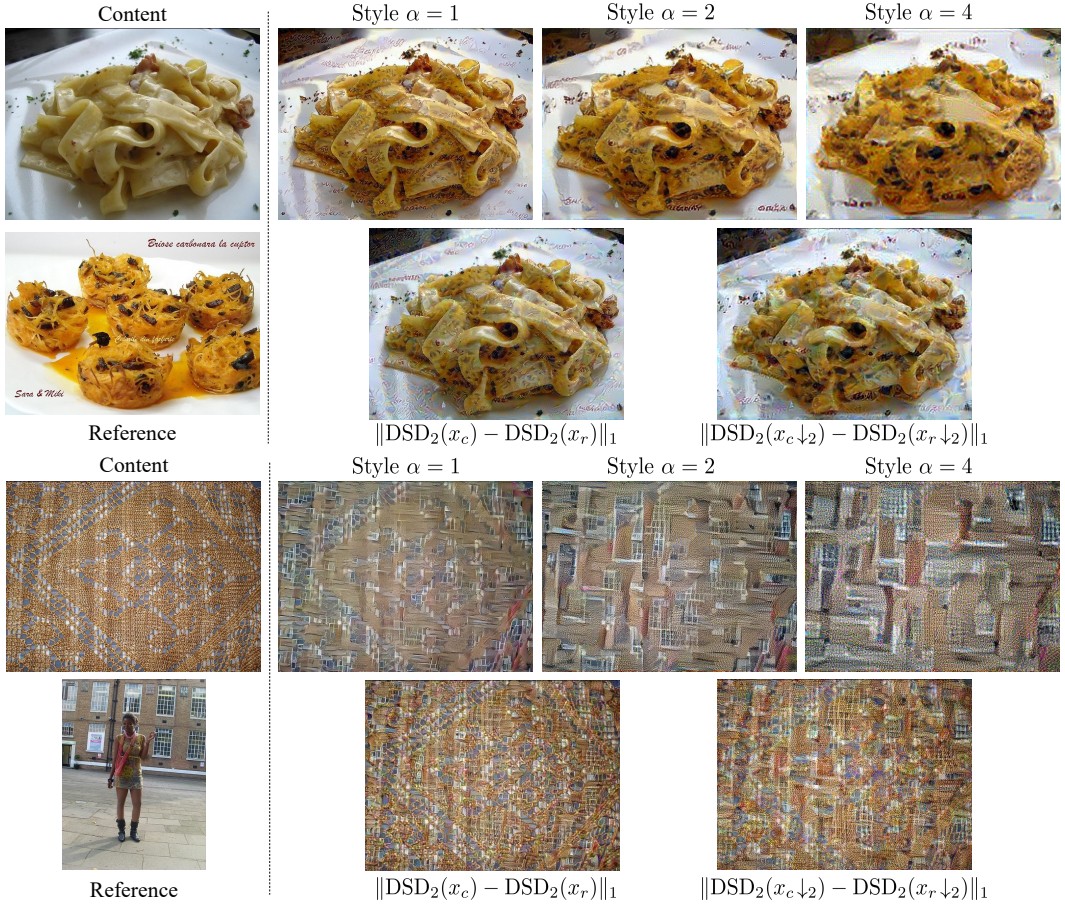

Figure 11: **Visualizing DSD.** To visualize what the DSD fingerprint captures, we modify the content image to have a similar fingerprint to that of the reference image (second and fourth rows). We do this for two pairs of scales (between full res. and $\alpha = 2$, and between $\alpha = 2$ and $\alpha = 4$). For comparison, we also perform the same experiment using the style loss for the three scales (first and third rows). As can be seen, matching DSD fingerprints requires modifying finer details.

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

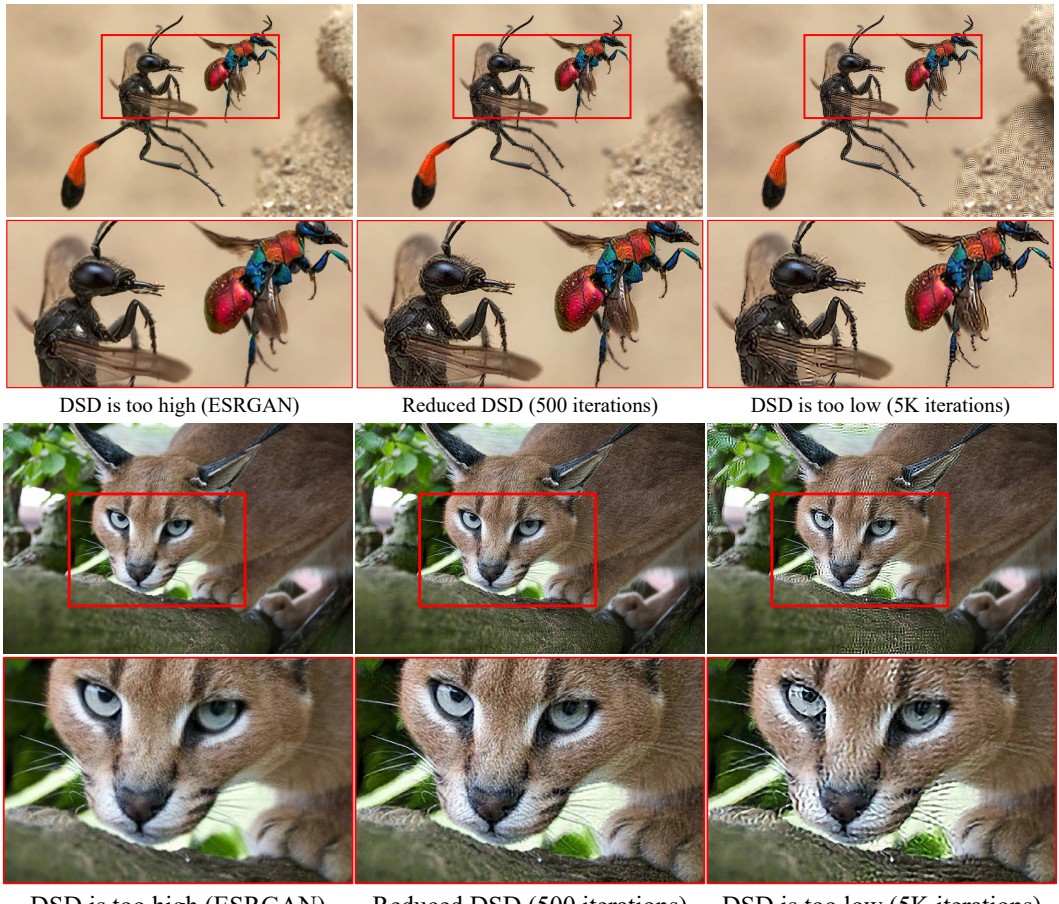



DSD is too high (ESRGAN)    Reduced DSD (500 iterations)    DSD is too low (5K iterations)



Figure 12: **Minimizing deep self-dissimilarity.** We minimize $\|\mathrm{DSD}_4(x)\|_1$ starting from an output by the ESRGAN [6] method (left). The result (right) after 5K iterations contains undesired artifacts. However, stopping the minimization earlier (after 500 steps) yields a sharper and more natural looking image (middle), compared to the original ESRGAN output.

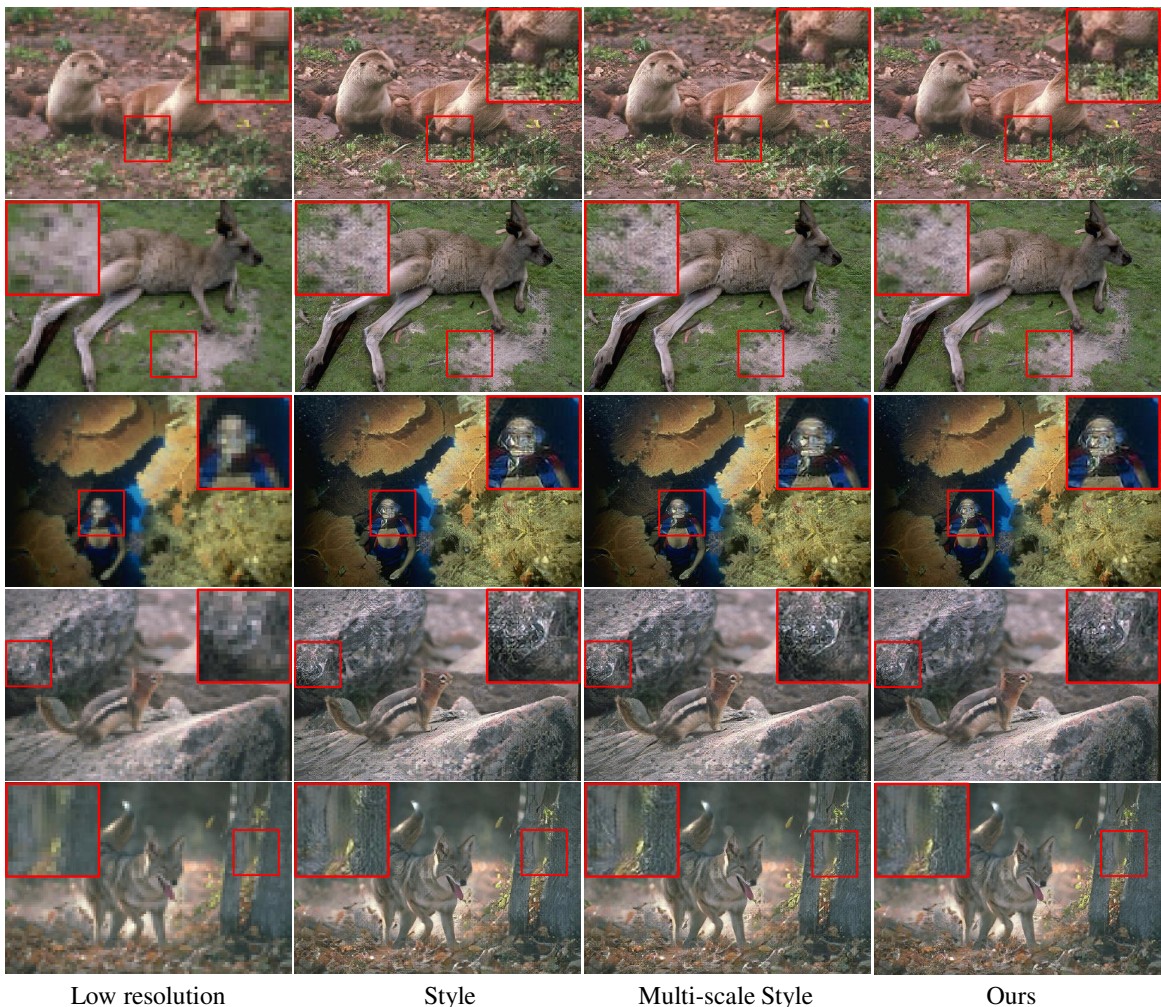

| Low resolution | Style | Multi-scale Style | Ours |

Figure 13: **Looking closer into the effect of DSD.** Substituting $\mathcal{L}_{\text{DSD}}$ with other loss terms that are based on feature distributions results in undesired grid artifacts (middle images). These artifacts get significantly reduced when using the DSD loss term (right). Here, all methods share the same light-weight network architecture which consists of only 5 residual blocks.

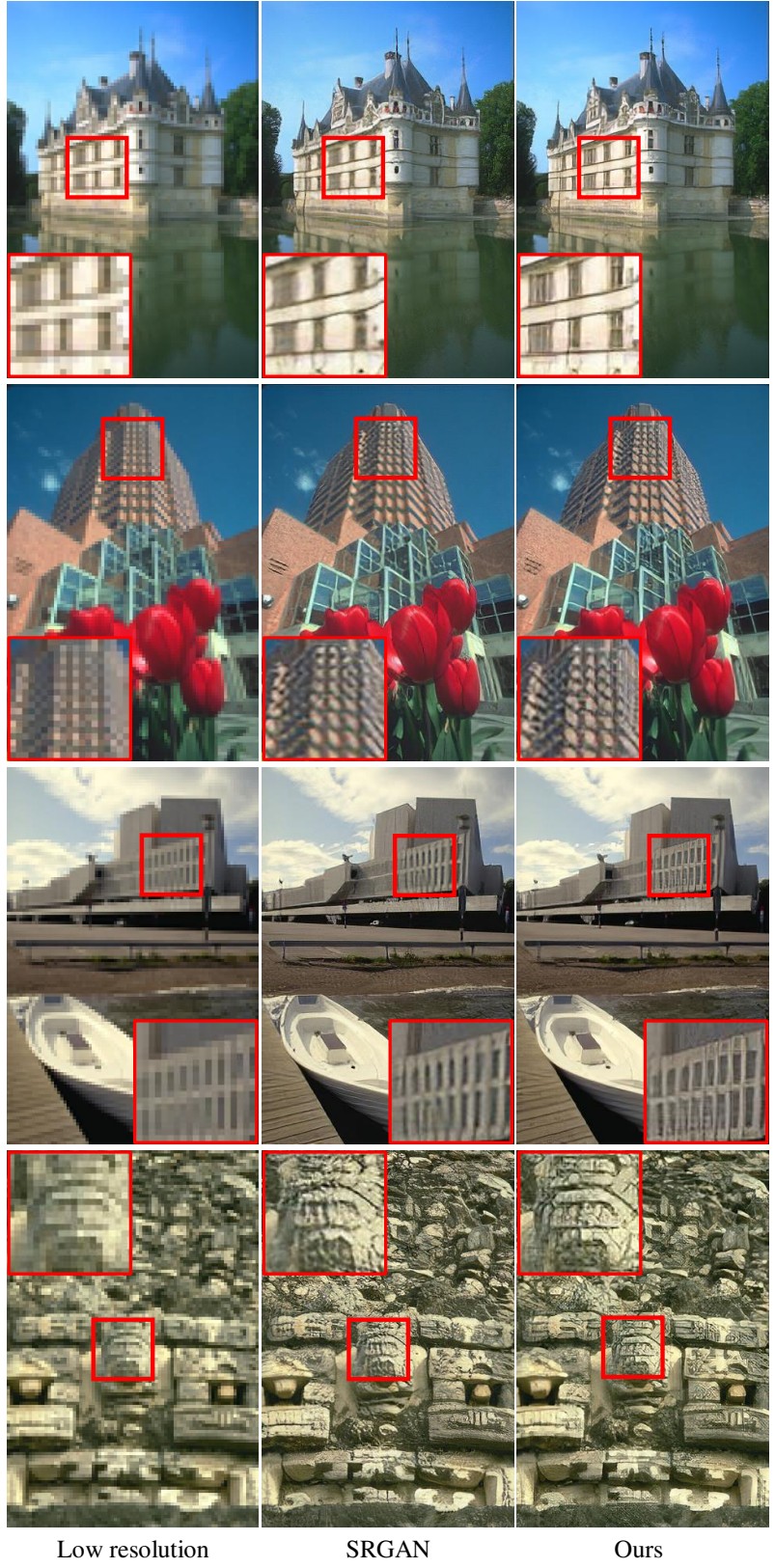

Low resolution         SRGAN         Ours

Figure 14: **Additional super-resolution comparisons.** We compare our method to the state-of-the-art SRGAN [4]. As is shown, our approach leads to sharper and more photo-realistic images, although we use no adversarial training and our model has fewer parameters (1.1M in our model and 1.54M in SRGAN).

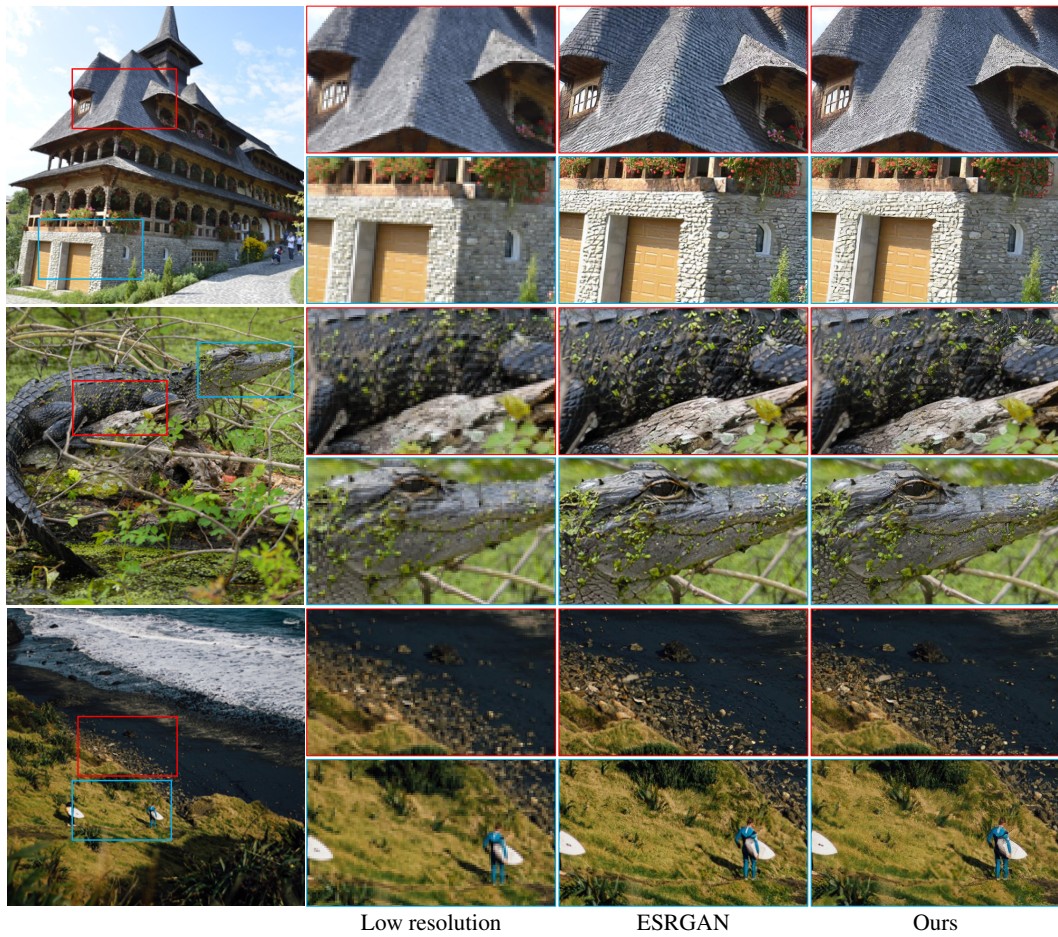

Low resolution        ESRGAN        Ours

Figure 15: **Additional super-resolution comparisons.** We compare our method to the state-of-the-art ESRGAN [6]. As can be seen, our approach presents competitive restorations comparing to ESRGAN, while avoiding adversarial training and using a model with only 6.5% the number of parameters of ESRGAN (1.1M for ours and 16.7M for ESRGAN).

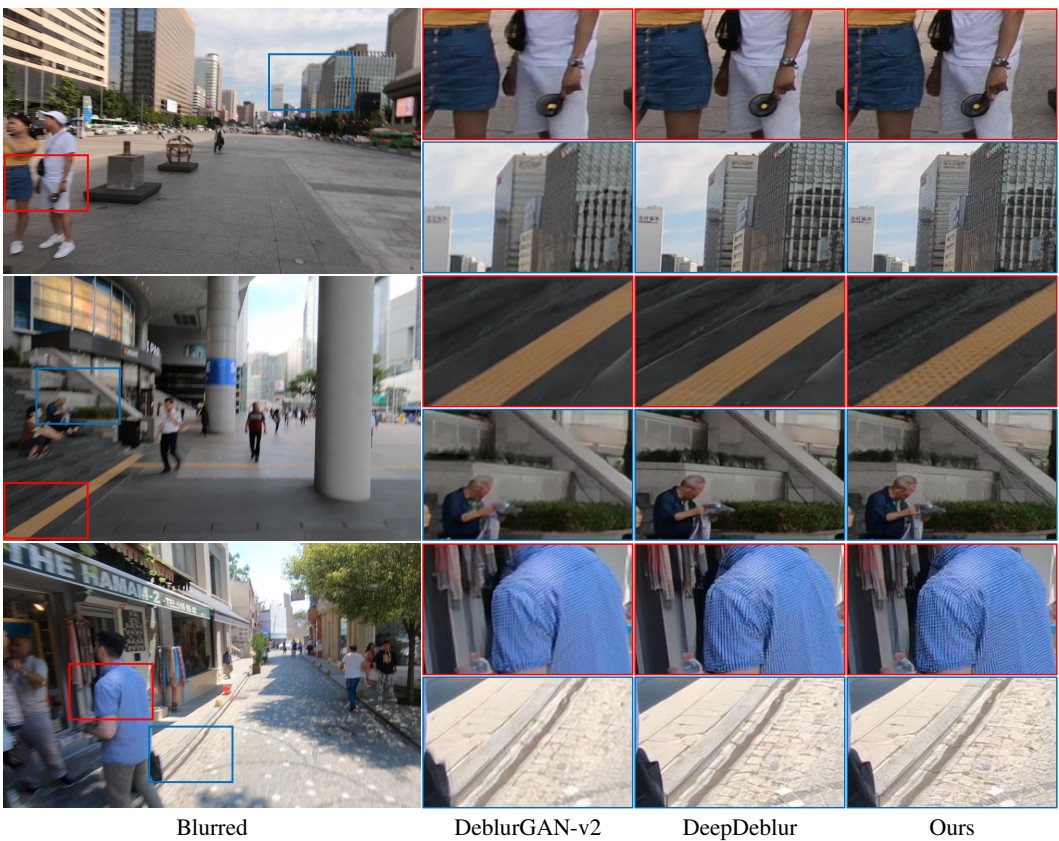

| Blurred | DeblurGAN-v2 | DeepDeblur | Ours |

Figure 16: **Additional motion-deblurring comparisons.** We visually compare our results with DeepDeblur [5] and DeblurGAN-v2 [3], two methods that utilize adversarial training. As can be seen, DSD achieves better visual quality with sharper, more natural appearing result, without using any adversarial training. Note also that our model is smaller than the competing methods, containing only 0.81M parameters, compared to 11.72M and 60.93M of DeepDeblur and DeblurGAN-v2, respectively.