# OpenReview forum: "Deep Self-Dissimilarities as Powerful Visual Fingerprints"
_NeurIPS.cc/2021/Conference — NeurIPS 2021 Spotlight_

### Official Review · Reviewer_C5ui · 2021-07-15

**Rating:** 6
**Confidence:** 4

**Summary:**

The authors observe that Gram matrices of image patches and of features extracted from convolutional networks behave differently. Namely that the latter are not invariant to image scale changes. Taking advantage of this invariance, they propose an image descriptor computed as a difference of Gram matrices of convnet features at two different image scales. The authors show the obtained descriptors can be used to compute an image quality measure in a full-reference setup that is in good alignment with human preference. An alternative no-reference image quality measure is also formulated and show cased in experiments for single-image super-resolution and motion debluring with promising results.

**Limitations And Societal Impact:**

Possible improvements:
- Fig. 2 - the caption should be improved, currently it does not give enough details to understand what is plotted, even though this information is in the main text. Consider labeling the x-axis of the plots clearer or mentioning in the caption (a) shows VGG layers and (b) patch sizes. Consider labeling the green bars as "Same image" instead of "Dissimilarity". And also consider replacing the y-label with "L_1 distance".
- Fig. 2 should include error bars.
- The definition (2) omits the layer from which the features are used. This makes it harder than necessary to think about the statements like the one on lines 91-92.
- In Table 1 I would expect to see results computed on the test set, however, the results for LPIPS are different from what was reported before. Make it clear which data is used to compute the results in Table 1.
- Connected to LPIPS results in Table 1, it is not clear which network was used for it, i.e., is this LPIPS-VGG,  LPIPS-Alex, LPIPS-Squeeze?
- Why does the Table 1 miss PieAPP results?
- The main text should be made more self-contained, definitely add details on the training/experiments setup, e.g., which resnet is meant on line 154, what the training setup is to allow for training a resnet with just 400 images, etc.
- Check that DeepDeblur and DeblurGAN-v2 are not mixed up in Fig. 8 (b).

Questions:
- Why is Fig. 2 showing the current selection of vgg layers, missing any layer deeper than 3-1?
- In Table 2, DSD4 performs good but DSD2 significantly worse, compared to this Table 1 shows values only for DSD2. What was the reasoning to report one or the other scale?
- Along the same lines as the point above, how is it selected which of the VGG layers is used to compute DSD?
- Were any other backbones than VGG tested?

**Main Review:**

The proposed method is simple yet could be interesting, however, understanding the paper is harder than necessary, some choices seem to be ad-hoc and reporting of the results seems to be selective. See the suggestions below.

UPDATE: Thanks to the authors for addressing the concerns raised. After adapting the text as discussed, I would consider the paper to be acceptable.

**Time Spent Reviewing:**

3

---

> ### Author Response · Authors · 2021-08-10
> **Thanks for the constructive feedback**
>
> **Figure 2:**
>
> Thank you for your feedback, we will change the caption to be more clear and will include error bars. Note that we plotted in the figure only the layers which we used in the experiments. But the behavior is qualitatively similar also for deeper layers. For example, for ReLU41 we have $4.195\cdot {10^{-3}}, 3.615\cdot {10^{-3}}$ and $5.530\cdot {10^{-3}}$ for random-images (blue), same-class (red) and self-dissimilarity (green), respectively. Therefore, in this layer too, the self-dissimilarity of images is at least as large as the dissimilarity between different images.
>
>
> **LPIPS scores in Table 1:**
>
> We did compute results on the PieAPP test set. We used the official pytorch implementation of LPIPS, version 0.1, with default setting of “net=alex'', as it is reported to achieve the best scores. We obtained a Pearson correlation coefficient of 0.641, whereas in the PieAPP webpage, the coefficient is reported to be 0.644. This small difference may be a result of different LPIPS versions or different frameworks (e.g. pytorch vs. TensorFlow).
>
>
> **including PieAPP results in Table 1:**
>
> Thank you for the suggestion. Note that rows 1-5 in Table 1 are generic measures, which were not trained on any specific dataset of degradations, or with any supervision of human mean opinion scores. We will add another table, comparing methods that were trained (or fine-tuned) on the PieAPP training set. This will include the PieAPP network, as well as finetuned versions of LPIPS and all other competitors in Table 1. Naturally, since the degradations in the PieAPP test set are similar to those in the training set, all such fine-tuned methods will achieve higher correlations with the human scores.
>
>
> **Definition (2) omits the layer from which the features are used, lines 91-92:**
>
> In lines 91-92 we use the layer ReLU31. Thank you for pointing this out. As we mentioned, we omit the layer from the DSD notation for keeping the paper concise, however following your suggestion we will include it back. We will also make sure the used layer is always mentioned.
>
> **Details on the training/experiments setup:**
>
> All the details regarding all the architecture we used and the exact training setting appear in the supplementary material. Please note that the setting of using 400 images for training a restoration network is common practice (see for example [R1] below).
>
> [R1] Zhang, Kai, et al. "Beyond a Gaussian denoiser: Residual learning of deep cnn for image denoising." IEEE transactions on image processing, 2017
>
>
>
> **DeepDeblur and DeblurGAN-v2 are mixed up?**
>
> Thank you. There is no mix here. We used a newer official version of DeepDeblur and the official implementation of DeblurGAN-v2. This is why DeepDeblur achieves lower NIQE and higher SSIM than DeblurGAN-v2. We’ll clarify this.
>
>
> **DSD4 and DSD2 in Table 1 and In Table 2:**
>
> We will add DSD$_4$ to Table 1, thanks. The results are:
>
> DSD$_4$:
>
> ReLU$_{21}$ - 0.570
>
> ReLU$_{31}$ - 0.667
>
> ReLU$_{41}$ - 0.607
>
> ReLU$_{51}$ - 0.600
>
> As stated in lines 152-153, in the no-ref setting (Table 2) we train a regression network under the assumption that we can estimate the DSD value of the clean image $x$ from a downsampled version of the degraded image, $y \downarrow _ \alpha$, as for many types of degradations $y \downarrow _ \alpha$ and $x \downarrow _ \alpha$ are quite similar. This assumption is better satisfied for stronger downsampling factors and this is the reason that DSD$_4$ performs better here.
> The full-reference case (Table 1), is a completely different setting and hence DSD$_4$ works slightly worse there.
>
> **how is it selected which of the VGG layers is used to compute DSD?**
>
> As described in line 170, in all image restoration experiments we used a combination of  {ReLU_21, ReLU_22, ReLU_31} activations, which were found empirically to work best. Our intuition is that using multiple layers enhances the sensitivity of DSD to multi-scale features (according to the receptive field of each layer). We also refer the reviewer to the ablation study in Table 3 that assesses the contribution of each of these activations to the SR performance.

---

> > ### Comment · Reviewer_C5ui · 2021-08-17
> > **Thanks**
> >
> > Thanks for your answers.

---

### Official Review · Reviewer_ooMi · 2021-07-15

**Rating:** 6
**Confidence:** 4

**Summary:**

This paper proposes the use of "deep self-dissimilarities" (DSD) - meaning differences in feature activations at deeper layers of a network when computed on an image versus on a downsampled version of that image.  DSD is applied to several tasks - serving as a measure for predicting human preference of image quality/distortion, used as a loss-term for single image super-resolution, and as a loss term for motion deblurring.  In all cases, the proposed method performs well relative to existing state-of-the-art.

**Main Review:**

This paper begins by making the interesting observation that feature activations in later stages of a deep network do not exhibit consistency/similarity when computed over different scales/downsamplings of an image.  This is in contrast to raw image patches or early feature activations of the network.  Therefore, this paper proposes using this difference in activations, "deep self-dissimilarity" (DSD), as a visual fingerprint, and apply it to several tasks.

First, DSD is used as a full reference distortion measure, as well as a no-reference quality measure, and in both cases is shown to correlate better to human preferences than existing methods.  The latter requires making an assumption that, given a degraded version y of an image x, DSD(X) can be well-predicted by DSD(y|alpha) where y|alpha is a version of y downsampled by a factor of alpha.  As a note, it would be interesting to have be given some numbers/sense of how well this assumption holds (eg the test accuracy of the regression network).

Second, DSD is used as an additional loss term for single image super-resolution SR and motion deblurring.  In both cases, DSD performs well relative to existing state-of-the-art, and, unlike many methods for SR, without requiring GAN training.

Overall, I felt that the paper was nice, in terms of making an interesting observation regarding deep self-dissimilarity, and based on this observation, proposing a novel measure that is empirically validated to give good performance on several tasks.

However, as a primary negative, from the paper, I don't have a strong sense or intuition for why the method works well.  What is the measure really keying in on, and what are possible failure cases/error modes?  Some explanation is given in section 3.3, but the overall argument seems to be quite high-level - that DSD focuses on smaller details compared with something like style dissimilarity.  A natural question that I don't think was addressed by the paper is how much the method depends on the specifics of the network from which the DSD is being computed?  For instance, how would the results change if that underlying network was trained with much more aggressive data augmentation or some architectural change to make it more scale-invariant?  Or, in another direction, Figure 2 gives mean dissimilarity, but how much variance is there?  Are there certain classes of images for which DSD is more/less informative?

**Time Spent Reviewing:**

3

---

> ### Author Response · Authors · 2021-08-10
> **Thanks for the very helpful comments and great ideas**
>
> **DSD(X) is predicted by DSD(y|alpha), how well does this assumption hold?**
>
> This is indeed a very interesting point to explore. The test RMSE of the regression network is $1.21\cdot 10^{-3}$ for ReLU$_{31}$ and $\alpha = 4$. This is lower than the standard deviation of DSD(x) over the same test set, which is given by $3.31\cdot 10^{-3}$ (the standard deviation is the RMSE that would be achieved by a naive predictor that always predicts the average DSD value of natural images, regardless of the input y|alpha).
>
> **What is the measure really keying in on? Are there certain classes of images for which DSD is more/less informative?**
>
> This is an important point, which we tried to explore through some visualizations in the paper and SM, but we agree it would be beneficial to further analyze through more experiments.
> First, note that the experiments in Fig. 5 (and Fig. 12 in the SM) shed some light on this: By modifying an image so as to minimize the $\ell_1$ norm of its DSD (i.e. maximizing the deep feature *similarity* across scale), we get images with unnatural repetitive textures. These are images with significant deep similarity across scales.
> Another interesting experiment is to repeat this, while keeping the “downsampled image” for the DSD fixed. When performing such experiments we observed that minimal DSD is obtained when the full-res image contains repetitive structures from the LR image. We will add these results to the SM.
> These experiments support our observation that images with a small DSD typically don’t look natural. A complementary question is what are the natural images that have smaller DSDs than others.
> To answer this, we can examine the images with the highest/lowest DSD values within some dataset. When performing this experiment on the dataset used in Fig. 2, we observed that natural images with the lowest DSD tend to include large smooth areas, whereas images with the largest DSD contain high frequency textures. This is to be expected, as the high frequencies are lost when down-sampling the image, which may completely modify the distribution of deep features. We observe a similar behavior when examining certain classes within the ImageNet validation set. For example, “sandbar” has a DSD of $2.1\cdot 10^{-3} \pm 0.83\cdot 10^{-3}$, whereas “window screen” has $13.5\cdot 10^{-3} \pm 11.0\cdot 10^{-3}$ .
> We will add these experiments to the SM.
>
> **How would the results change if the underlying network was trained to be scale-invariant?**
>
> Interesting point. We will explore this point by training VGG with much more aggressive scaling augmentation as suggested, and will add this to the discussion of the paper. We anticipate: (1) this will reduce the classification accuracy of VGG, as the 224x224 images in ImageNet contain objects of typical sizes, which the network often exploits, (2) our DSD measure will be much less informative as the distribution of deep features will experience much smaller between-scale variability.
>
> **Figure 2 gives mean dissimilarity, but how much variance is there?**
>
> We will add error bars to this figure. Generally we observed that the variance is quite large (as shown in the histogram of fig. 5),  so the differences between the mean dissimilarities of deep layers are statistically insignificant. That is, the degree of self-dissimilarity is not statistically different from the degree of dissimilarity between different images. We will also include a normalized version of this plot with each of the calculated image distances ($d(x,y)$, with mean value presented in the red and blue bars) normalized by its corresponding self-similarity measure ($d(x,x^\alpha)$, with mean value presented in the green bar).

---

### Official Review · Reviewer_CTwK · 2021-07-16

**Rating:** 7
**Confidence:** 3

**Summary:**

This paper proposes a new image descriptor, called DSD (Deep Self-Dissimilarities). The authors observe that the very same image may happen to be classified in different ways by deep neural networks when viewed at different scales. For example (Figure 1) an image is classified by the same network as "Jean" at full resolution and as "Constrictor" at a lower scale. Therefore, to characterize an image, they consider the activations at a given layer of the net for the original image and its resized version, and compute some statistics that describe how they change. More precisely, the descriptor is computed as the difference between the Gram matrices of activations at full and reduced scale. Several experiments are carried out to illustrate the potential of the proposed descriptor.


**Limitations And Societal Impact:**

The authors did not describe limitations and potential negative societal impacts of their work. For what concern limitations it would be interesting to better understand why for the image super-resolution and motion deblurring tasks, the proposed descriptor is not very effective.


**Main Review:**

The paper moves from some interesting considerations. In fact, the very same network makes different decisions for the same image depending on the image scale, and these variations may be descriptive of the image by themselves. The proposed descriptor seems to be able to capture such variations. However, using deep features to characterize images is hardly new, as the authors themselves acknowledge. So prior work on this topic should be analyzed adequately (just a few lines in the Related Work now) and similarities and differences with other proposal should be clearly pointed out. This would help putting the proposal in a correct perspective with respect to the state of the art. Along this line, it would be interesting to investigate in more depth what these observed dissimilarities depend upon, that is, which image features appear to be invariant/variant across scales.

Experiments on the possible uses of DSD provide mixed support for this proposal. Those concerning its use as full-reference and no-reference quality measures show an impressive performance and maybe should be expanded. The performance observed for image super-resolution and motion deblurring instead is less convincing and not above the state of the art. This is confirmed by the selected visual examples, Figure 3, 7, 9 and 10 where, contrary to what suggested by the comments, hardly any improvement can be noticed with respect to the references.

In summary, DSD is an interesting descriptor, with good potential, but should be better positioned with respect to the relevant literature and more deeply analyzed and justified.

Minor
- Eq.(2): I believe some constant may be missing to make the two terms, computed on image of different size, comparable
- Figure 2: what does rx_y stands for?
- typos: debluring

-------------------------------------------------------------------------------------------------------------------------------------------------------------------------------

UPDATES

First of all, I want to thank the authors for answering my questions.
I appreciate the considerations about which features are invariant across image scales.
The analysis of the images with the highest and lowest DSD values is very interesting and can be of benefit for the paper. I also better analyzed the visual results for super-resolution and motion deblurring and I can see some improvements,
however I believe that a pointer in the text would definitely help.
Finally, for what concerns the prior work, I don't think the paper needs additional references or comparisons, but just an expansion of the related work analysis made in Section 2. This would allow a non-expert reader to better position the work with respect to prior art.

Overall, the authors addressed my major concerns and for this reason I improve my final rating.

**Time Spent Reviewing:**

4

---

> ### Author Response · Authors · 2021-08-10
> **Thank you for the constructive feedback**
>
>
> **Covering additional deep feature based methods:**
>
> We completely agree that deep feature descriptors are in widespread use. For keeping the paper concise and self contained, we cover only deep feature measures that are used for image quality assessment, and for image restoration, which are at the focus of our paper. We emphasize the difference between our measure and commonly used measures in the introduction (third paragraph) and in Fig. 4. We also perform extensive comparisons to all state-of-the-art deep-feature based losses that we are aware of, including the perceptual loss, the style loss (and multi-scale variants we added), the projected distribution loss, and the contextual loss (CX). Please see Table 1, Table 3, Fig. 8 (blue points), Fig. 9, and Fig. 10. We will be happy to cover additional related works and to compare to any additional deep-feature based loss the reviewer can point us to.
>
> **Which image features appear to be invariant/variant across image scales?**
>
> This is an excellent point. First, note that the experiments in Fig. 5 (and Fig. 12 in the SM) shed some light on this: By modifying an image so as to minimize the $\ell_1$ norm of its DSD (i.e. maximizing the deep feature *similarity* across scale), we get images with unnatural repetitive textures. These are images with significant deep similarity across scales.
> Another interesting experiment is to repeat this, while keeping the “downsampled image” for the DSD fixed. When performing such experiments we observed that minimal DSD is obtained when the full-res image contains repetitive structures from the LR image. We will add these results to the SM.
> These experiments support our observation that images with a small DSD typically don’t look natural. A complementary question is what are the natural images that have smaller DSDs than others.
> To answer this, we can examine the images with the highest/lowest DSD values within some dataset. When performing this experiment on the dataset used in Fig. 2 (ImageNet validation set), we observed that natural images with the lowest DSD tend to include large smooth areas (see for example ILSVRC2012_val_00030176, ILSVRC2012_val_00025936, ILSVRC2012_val_00021306), whereas images with the largest DSD contain high frequency textures (see for example ILSVRC2012_val_00007953, ILSVRC2012_val_00008607, ILSVRC2012_val_00025873). This is to be expected, as the high frequencies are lost when down-sampling the image, which may completely modify the distribution of deep features. We will add this experiment to the SM.
>
>
> **The performance observed for image super-resolution and motion deblurring is less convincing:**
>
> We believe our results are significantly better than existing baselines. For example, in Fig. 10, the pattern on the wall in our reconstruction is much sharper than with DeblurGAN, and there are no halos around the hand in our method. Similarly, in Fig. 7, the fur and rock are significantly sharper in our reconstruction. We will add these pointers to the text. Similar improvements are also observed in the many additional results included in the SM. All these can be better observed when zooming into the images. This is also supported by our quantitative evaluations (see Fig. 8).
> Additionally, please note that we are the first to achieve high perceptual quality in SR and blind deblurring *without* a GAN loss (which is usually very unstable to train).
> Finally, as we describe in the SM, we achieve all these while using a much lighter architecture; e.g. for the task of SR we have only $1.1$M parameters vs. $16.7$M parameters used for the ESRGAN model, and for blind-debluring we have only $0.81$M parameters vs. $60.93$M parameters used for DeblurGAN-v2. We’ll better highlight this point in the final version. Thus, we believe our experiments establish that our model does have a significant advantage over the baselines.
>
> **Eq.(2): constant is missing**
> Thanks for catching this. Eq. (1) should correspond to an average, not sum. This makes G (and as a result also Eq. (2)) invariant to the image size. We’ll correct this for the final version.
>
> **Figure 2: what does rx_y stand for?**
> This refers to the activation layer taken for calculating DSD (e.g. ReLU$_{xy}$ as written in the text). We will correct this notation and will make sure to clearly describe it in the text.

---

### Official Review · Reviewer_n5Q1 · 2021-07-16

**Rating:** 8
**Confidence:** 5

**Summary:**

The authors introduce a measure, which they deem the deep-self-dissimilarity (DSD), which measures the dissimilarity between deep-features from the same image presented at different scales.  They show that this can be used as a powerful visual fingerprint, and can be adapted to both a full-reference and no-reference image quality metric.

**Main Review:**

This is a clever and novel result, as far as I can tell, no other authors have suggested a measure of scale-wise dissimilarity of deep features as a descriptive measure of images, or one that has practical uses. The key idea, that despite images being self-similar in different scales in the pixel space, they are in fact quite dissimilar in different scales in deep feature response space, is also very novel and interesting on its own.

The performance of the full-reference metric on the pie-app dataset (outperforming LPIPS) is itself very interesting and impressive.

The visual results from the images are also impressive and suggest the measure is aligned with perception about images, though it would be fascinating to understand better how and why it aligns with perception.

A few comments:

I disagree with the description of it as a no-reference metric.  It is more accurately described as a reduced-reference quality metric.  Though it is true that the full scale reference image is not available to it, extracted statistics from a version of the image are available, which is more accurately reduced-reference. For the test case of super-resolution, it just turns out the degradation also coincides with the reduction to the reference.  This would not hold necessarily for other tasks requiring a no-reference metric.

Additionally, if you optimized a super resolution network just for the DSD metric, the output images will necessarily be related to the input images given no other constraints unlike other no-reference metrics which are comparisons between the output images and a distribution of natural images only, that will be optimized by output images that have no relation to the inputs.  This is because the DSD metric is actually a reduced-reference that also depends on the input, which is a reduced version of the reference.

This doesn't diminish the usefulness or novelty of the measure, especially in the case of super-resolution, but it should be clarified.


I'm not sure that I understand the experiment being run in figure 3 or what purpose it serves for the larger argument of the paper.



**Time Spent Reviewing:**

2

---

> ### Author Response · Authors · 2021-08-10
> **Thanks for the insightful observations**
>
> **No-reference metric:**
>
> The measure described in Sec. 3.5 is a no-reference metric in the sense that when used to measure the quality of an image $y$, it depends only on $y$ (see Eq. (5)). It has no access to any “reference” image $x$, not even to a down-scaled version of $x$. This is in contrast to the full-reference metric described in Sec. 3.4 (Eq. (4)), which does use $x$ in addition to $y$. Perhaps the confusion is rooted in the description of our logic behind the no-reference measure, which is similar to the full-reference metric of Sec. 3.4. But note that in the no-reference case, instead of using the unavailable reference image $x$ (or any version of it), we use a downscaled version of the available input image $y$ (note that this is true for all types of degradations, not only SR). We’ll do our best to clarify this in the final version.
>
>
> **Best suited for SR, less for other no-ref. tasks:**
>
> Note that the PieAPP dataset includes a variety of distortions besides SR (additive Gaussian noise, speckle noise, non-eccentricity, contrast sensitivity, deblurring, denoising, compression, geometric transformations, color transformations). Our no-reference metric is very effective in quantifying the degree of degradation in all those images, outperforming other no-reference methods (Table 2).
>
>
> **Figure 3:**
>
> The motivation behind the experiment shown in figure 3 is to visualize what DSD captures. We modify an image so as to minimize its distance to a reference image across different scales, using a style distance and using a DSD distance. Contrary to constraining style at lower scales, which reproduces a blurry image, matching DSD fingerprints requires modifying finer details.

---

> > ### Comment · Reviewer_n5Q1 · 2021-08-10
> > **Thank you for the clarification**
> >
> >
> > # No reference metric and super-res comments.
> > I believe any confusion that arose was in the exact language around degradation/downsampling.  I came away from reading that section with the understanding that the degradation was downsampling (which would make the degraded version of the image simply a reduced reference).  What you actually used was a degraded version of the image that was also then downsampled.  Clarifying some of that language should help.  I also was under the impression that it was the version of DSD you used to train the super-res network in the next section, but upon closer reading realized I was mistaken.

---

### Decision · Program_Chairs · 2021-09-27

**Decision:**

Accept (Spotlight)

**Comment:**

This paper addresses a new image descriptor, referred to as deep self dissimilarity (DSD) which measures the dissimilarity between deep-features from the same image presented at different scales. Reviewers agree that the paper presents a novel idea which suggests a  measure of scale-wise dissimilarity of deep features as a descriptive measure of images. The paper is well written and the key idea is appreciated. The rebuttal addressed most of concerns raised by reviewers, leading that two of reviewers raised the score during the discussion period. I believe that the paper is deserved to be presented in the conference.